# Nesterov Method for Asynchronous Pipeline Parallel Optimization

**Thalaiyasingam Ajanthan** [1]   **Sameera Ramasinghe** [1]   **Yan Zuo** [1]   **Gil Avraham** [1]   **Alexander Long** [1]

## Abstract

Pipeline Parallelism (PP) enables large neural network training on small, interconnected devices by splitting the model into multiple stages. To maximize pipeline utilization, asynchronous optimization is appealing as it offers 100% pipeline utilization by construction. However, it is inherently challenging as the weights and gradients are no longer synchronized, leading to *stale (or delayed) gradients*. To alleviate this, we introduce a variant of Nesterov Accelerated Gradient (NAG) for asynchronous optimization in PP. Specifically, we modify the look-ahead step in NAG to effectively address the staleness in gradients. We theoretically prove that our approach converges at a sublinear rate in the presence of fixed delay in gradients. Our experiments on large-scale language modelling tasks using decoder-only architectures with up to **1B parameters**, demonstrate that our approach significantly outperforms existing asynchronous methods, even surpassing the synchronous baseline.[†]

## 1. Introduction

Pipeline Parallelism (PP) is a standard parallelism technique for training large-scale frontier foundational models (Dubey et al., 2024; Liu et al., 2024a). PP partitions neural networks into sequential stages, enabling the training of models that exceed the memory capacity of any single device by distributing computations across multiple interconnected devices. The devices can be co-located in datacenters or connected via the internet (*i.e.*, low bandwidth connections) in a fully decentralised setting. Each device processes a stage (*i.e.*, a set of consecutive layers) and communicates the activations and gradients with adjacent stages often via bandwidth-constrained interconnects.

The main objective of PP methods is to mask the communication overhead and improve device utilization so as to minimize training time. To this end, many pipeline scheduling strategies (*i.e.*, the order of processing forward and backward passes of microbatches) have been developed (Huang et al., 2019; Narayanan et al., 2021b; Qi et al., 2023). However, the main bottleneck in these methods is the requirement to synchronize the weights and gradients across stages at each update step, hindering 100% pipeline utilization.

To enable full pipeline utilization, we consider asynchronous optimization in PP, where the weight updates are performed independently at each stage without waiting for the corresponding backward pass to complete. This approach introduces *gradient staleness*, as weights are updated multiple times between the forward and backward passes of a microbatch, resulting in outdated gradients being used for weight updates. This gradient staleness presents a significant optimization challenge, necessitating sophisticated delay correction mechanisms to ensure convergence, even in traditional distributed settings (Agarwal & Duchi, 2011; Zheng et al., 2017; Stich & Karimireddy, 2019; Mishchenko et al., 2022).

Typically, delay correction methods directly estimate the gradients via forecasting approaches (Zheng et al., 2017; Liu et al., 2024b) or extrapolate the previous update step in the weight space (Hakimi et al., 2019; Guan et al., 2019). We follow the latter approach as *delay correction in the weight space* does not make any assumptions on the loss landscape (as in gradient forecasting methods), but rather assumes that the update directions change slowly. Noting that the smoothness of update directions can be controlled in momentum based optimizers, we derive a look-ahead based optimization method to alleviate gradient staleness.

Specifically, we introduce a variant of the Nesterov Accelerated Gradient (NAG) method (Nesterov, 1983; 2013) for asynchronous PP optimization. Our idea stems from the observation that NAG has a look-ahead step which can be repurposed as a delay correction in the weight space, by carefully modifying the update formula (refer to Fig. 1). Our approach is a simple, yet elegant modification to NAG that does not introduce any hyperparameters. We theoretically prove that our approach converges at a sublinear rate for convex, smooth functions with a fixed delay in gradients.

We demonstrate the merits of our approach on large-scale

---

[1]Pluralis Research. Correspondence to: Thalaiyasingam Ajanthan <aj@pluralis.ai>.

*Proceedings of the $42^{nd}$ International Conference on Machine Learning*, Vancouver, Canada. PMLR 267, 2025. Copyright 2025 by the author(s).

[†]Our code is available at https://github.com/PluralisResearch/AsyncPP.

language modelling tasks with decoder-only transformer architectures (Vaswani et al., 2017; Karpathy, 2022). In short, our approach significantly outperforms all existing asynchronous PP methods including sophisticated delay correction mechanisms, even surpassing the synchronous baseline. Notably, for the first time, we train a **1B parameter model** to convergence in the asynchronous PP setup outperforming the synchronous baseline. Moreover, we show the effectiveness of our method in a realistic decentralized training framework, namely SWARM (Ryabinin et al., 2023), where our approach significantly outperforms both synchronous and asynchronous methods. Our experiments clearly demonstrate the feasibility of asynchronous PP optimization in the large-scale setting.

Our contributions can be summarized as follows:

- For the first time, we show an asynchronous PP optimization method can surpass the synchronous alternative for large-scale language modelling tasks.
- Our approach is an elegant variant of NAG, and we provide theoretical and empirical justification of convergence in the presence of gradient staleness.
- Furthermore, we test our method in a realistic decentralized training framework (SWARM), proving its empirical effectiveness beyond doubt.

## 2. Preliminaries

We first define the problem setup and briefly review PipeDream (Narayanan et al., 2019) and NAG (Nesterov, 2013; Bubeck et al., 2015), upon which we build our work. We refer the interested reader to the respective papers for more details.

### 2.1. Problem Setup

Let us consider a single pipeline without data parallelism for simplicity. Let $P$ be the number of pipeline stages. Let $f_i(\mathbf{w}_i, \mathbf{x}_{i-1})$ be the forward function in stage $i$ with weights $\mathbf{w}_i$ (correspond to all learnable parameters) and input $\mathbf{x}_{i-1}$, respectively. We may simply write $f_i$ for brevity, when the context is clear. Now, the full neural network forward function can be written as:

$$F(\mathbf{W}, \mathbf{x}_0) := f_P \circ f_{P-1} \circ \cdots \circ f_1(\mathbf{x}_0), \qquad (1)$$

where $\mathbf{W} = \{\mathbf{w}_P, \ldots, \mathbf{w}_1\}$ and $\mathbf{x}_0$ is the input data point. Analogously the backward function can be written as:

$$G(\mathbf{W}, \mathbf{e}_P) := g_1 \circ g_2 \circ \cdots \circ g_P(\mathbf{e}_P), \qquad (2)$$

where $\mathbf{e}_P$ is the error signal, and $g_i(\mathbf{w}_i, \mathbf{e}_i)$ is the backward function at stage $i$ corresponding to $f_i$. Specifically, $\mathbf{e}_i$ is the gradient of the loss with respect to the output activations $\mathbf{x}_i = f_i(\mathbf{w}_i, \mathbf{x}_{i-1})$, and it is backpropagated through the network using the weights (and non-linearities) of each

stage. Formally, the gradient with respect to the weights and the error signal to the previous stage can be written as:

$$\nabla f_i(\mathbf{w}_i^t) = h_i(\mathbf{w}_i^t, \mathbf{e}_i^t), \qquad (3)$$
$$\mathbf{e}_{i-1}^t = g_i(\mathbf{w}_i^t, \mathbf{e}_i^t),$$

where $h_i$ corresponds to the chain rule. Then at each stage, the computed gradients $\nabla f_i(\mathbf{w}_i^t)$ are used to perform the optimization step with learning rate $\eta > 0$:

$$\mathbf{w}_i^{t+1} = \mathbf{w}_i^t - \eta \nabla f_i(\mathbf{w}_i^t). \qquad (4)$$

We omit optimizer specific updates and stochasticity for simplified notation.

Note that, in the asynchronous setting, the weights $\mathbf{w}_i^t$ and gradients $\mathbf{e}_i^t$ (and $\nabla f_i(\mathbf{w}_i^t)$) are not synchronized. In other words, the weights are updated without waiting for the backward passes (*i.e.*, gradient computation) of all active microbatches to complete, leading to delayed gradients. Therefore, asynchrony would affect both the backpropagation step, Eq. (3), and the optimization step, Eq. (4).

### 2.2. PipeDream

PipeDream (Narayanan et al., 2019) uses a One-Forward-One-Backward (1F1B) schedule with weight stashing for asynchronous PP optimization. At steady state, each stage alternates between forward and backward passes. Since *weight stashing* retains a copy of weights used in the forward pass until the microbatch completes, correct backpropagation can be computed by loading the stashed weights. This ensures synchronous backpropagation, Eq. (3), while weight updates, Eq. (4), remain asynchronous, *i.e.*, delayed gradients are used to update the most recent weights.

Precisely, let $\tau_i$ be the delay at stage $i$, meaning the weights at stage $i$ are updated $\tau_i$ times between the forward and backward passes of a particular microbatch. Assuming constant delay at each stage, this delay can be written as:

$$\tau_i = \left\lfloor \frac{2(P - i) + 1}{2K} \right\rfloor, \qquad (5)$$

where $i \in \{1, 2, \ldots, P\}$ and $K$ is the update interval, *e.g.*, $K = 1$ if updated for every microbatch. Note that earlier stages incur larger delays.

Now the backward function, and optimizer step at time $t$ for PipeDream can be written as:

$$\mathbf{e}_{i-1}^{t-\tau_i} = g_i(\mathbf{w}_i^{t-\tau_i}, \mathbf{e}_i^{t-\tau_i+1}), \qquad (6)$$
$$\mathbf{w}_i^{t+1} = \mathbf{w}_i^t - \eta \nabla f_i(\mathbf{w}_i^{t-\tau_i}).$$

To reiterate, the forward and the backward passes are computed on the weights at time step $t - \tau_i$, and these outdated gradients are used to update the latest weights at time step

$t$. In short, PipeDream uses more memory in each stage to store old weights, ensuring correct gradient computation, but the optimization is performed asynchronously without any delay correction.

Additionally, due to asynchrony, PipeDream does not guarantee that weights are synchronized across stages. Specifically, since the delay is stage dependent, earlier stages will use older weights compared to the later stages. Precisely, the forward function at time step $t$ takes the following form:

$$F(\mathbf{W}^t, \mathbf{x}_0) := f_P^t \circ f_{P-1}^{t-1} \circ \cdots \circ f_1^{t-P+1}(\mathbf{x}_0), \quad (7)$$

where $f_i^t := f_i(\mathbf{w}_i^t, \cdot)$. The gradients are computed for this function. Note, for $t \geq P$, all stages are updated at each time step following the 1F1B schedule. Thus, the set of weights $\mathbf{W}^P$ can be interpreted as stage dependent initialization, and in practice, this weight discrepancy has not shown to cause any convergence issues (Narayanan et al., 2019).

### 2.3. Nesterov Accelerated Gradient

Nesterov Accelerated Gradient (NAG) (Nesterov, 1983; 2013; Bubeck et al., 2015) is an accelerated gradient method that has the optimal $O(\frac{1}{t^2})$ convergence rate for smooth convex functions in the non-stochastic setting. The main idea is to perform a look-ahead step in the previous update direction, combined with a carefully selected sequence of step-sizes to ensure accelerated convergence.

Let $f : \mathbb{R}^m \to \mathbb{R}$ be the objective function. Then, NAG performs the following iterations starting from an initial point $\mathbf{w}_1 \in \mathbb{R}^m$:

$$\mathbf{d}_t = \gamma_t(\mathbf{w}_t - \mathbf{w}_{t-1}), \quad (8)$$
$$\mathbf{w}_{t+1} = \mathbf{w}_t + \mathbf{d}_t - \eta \nabla f(\mathbf{w}_t + \mathbf{d}_t),$$

where $\eta > 0$ is the learning rate. Here, the momentum coefficient $\gamma_t$ satisfies, $\gamma_1 = 0$, $0 < \gamma_t < 1$, and the sequence of $\gamma_t$ is derived as part of the convergence proof (Bubeck et al., 2015). Note that, $\mathbf{d}_t$ corresponds to the look-ahead step which extrapolates the update $(\mathbf{w}_t - \mathbf{w}_{t-1})$ by $\gamma_t$ and the gradients are computed at the extrapolated point $(\mathbf{w}_t + \mathbf{d}_t)$.

NAG has been incorporated into popular deep learning optimizers such as SGD (Sutskever et al., 2013) and Adam (Dozat, 2016), although it often slightly underperforms in synchronous settings. However, we show the superiority of a variant of NAG in asynchronous optimization.

## 3. Method

We address the gradient staleness in asynchronous PP optimization by incorporating a delay correction mechanism. Specifically, our idea is to do the *delay correction in the weight space* by extrapolating the last update step, so that the gradients can be computed at a point that is closer to

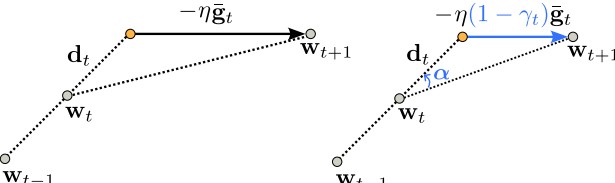

Figure 1: *Original NAG (left) and our modified version (right) for delayed gradients (denoted with $\bar{\mathbf{g}}_t$). Our method discounts the gradient term by $(1 - \gamma_t)$. When $\gamma_t \to 1$, the angle $\alpha \to 0$, making the weight trajectory smoother. Consequently, the look-ahead $\mathbf{d}_t$ can be shown to act as delay correction, alleviating gradient staleness.*

the ideal one. This is appealing as it does not make any assumptions about the loss function or gradients as in gradient forecasting methods. The only assumption is that the update directions change slowly with respect to iterations, which is valid for momentum based optimizers.

As discussed previously, NAG is an ideal candidate for this as it performs a look-ahead step by extrapolating the last update step. We now introduce our modified update formula for delay correction in the weight space.

### 3.1. Nesterov Method for Delayed Gradients

Let us consider a particular stage and drop the stage index for brevity.[1] Let $\tau$ be the delay as defined in Eq. (5), which is constant for each stage, and $\Delta_t$ be the corresponding delay in the weight space. Then, the delayed versions of $\mathbf{w}_t$ and $\mathbf{d}_t$ take the following form:

$$\bar{\mathbf{w}}_t = \mathbf{w}_{t-\tau} = \mathbf{w}_t - \Delta_t, \qquad \bar{\mathbf{d}}_t = \mathbf{d}_{t-\tau}. \quad (9)$$

Now, our variant of NAG with delayed gradients perform the following iterations:

$$\mathbf{d}_t = \gamma_t(\mathbf{w}_t - \mathbf{w}_{t-1}), \quad (10)$$
$$\mathbf{w}_{t+1} = \mathbf{w}_t + \mathbf{d}_t - \eta(1 - \gamma_t) \nabla f(\bar{\mathbf{w}}_t + \bar{\mathbf{d}}_t).$$

Note that compared to Eq. (8), our update formula discounts the gradient term by $(1 - \gamma_t)$. This is illustrated in Fig. 1. This subtle but important difference, allows us to show that our look-ahead step approximates $\Delta_t$ (*i.e.*, acts as delay correction) and our algorithm converges at a sublinear rate when the gradients are delayed.

**Look-ahead as delay correction.** The momentum coefficient $\gamma_t$ is usually chosen to be a constant close to 1 or chosen as an increasing sequence satisfying $\lim_{t\to\infty} \gamma_t = 1$. Assuming this, we can intuitively see that the influence of the gradient term decreases when $\gamma_t$ increases, due to the discount factor $(1 - \gamma_t)$. This translates into $\mathbf{d}_t$ dominating the updates, resulting in a smooth trajectory in the weight

---

[1]We also use a subscript for the time step to reduce clutter.

space. Consequently, it can be shown that the vector directions of look-ahead at time $t - \tau$, ie, $\bar{\mathbf{d}}_t = \mathbf{d}_{t-\tau}$, and the delay $\Delta_t$ are aligned.[2] This means that taking a step in the direction of $\bar{\mathbf{d}}_t$ reduces the delay, effectively acting as delay correction in the weight space. We formally state this below.

**Proposition 1.** Let $\gamma_t$ be an increasing sequence such that $\lim_{t \to \infty} \gamma_t = 1$ then, $\lim_{t \to \infty} \cos(\Delta_t, \bar{\mathbf{d}}_t) = 1$, where $\cos(\cdot, \cdot)$ is the cosine similarity.

*Proof.* By algebraic manipulations, we can write $\Delta_t$ in terms of $\bar{\mathbf{d}}_t$ as follows:

$$\Delta_t = \sum_{i=1}^{\tau} \left[ \left( \Pi_{j=t-\tau+1}^{t-i} \gamma_j \right) \bar{\mathbf{d}}_t \right. \tag{11}$$
$$\left. - \eta \sum_{k=t-\tau}^{t-i} \left( \Pi_{j=k+1}^{t-i} \gamma_j \right) (1 - \gamma_k) \bar{\mathbf{g}}_k \right],$$

where $\bar{\mathbf{g}}_t = \nabla f(\bar{\mathbf{w}}_t + \bar{\mathbf{d}}_t)$. When, $\gamma_t \to 1$, the gradient term vanishes, and $\cos(\Delta_t, \bar{\mathbf{d}}_t) \to 1$. Refer to Appendix A.2 for the detailed proof. $\square$

Here, since the last update step aligns with the delay direction, one may wonder what if we extrapolate in the direction of $(\mathbf{w}_t - \mathbf{w}_{t-1})$ to completely compensate for the delay rather than taking a fractional step $\mathbf{d}_t = \gamma_t(\mathbf{w}_t - \mathbf{w}_{t-1})$. The answer is, convergence may be disrupted if one naively extrapolates further than $\gamma_t$. Note that by definition of NAG, $\gamma_t$ cannot be larger than 1, and it is usually derived as part of the convergence proof. Nevertheless, our experiments show that our delay correction in the weight space is superior compared to gradient forecasting based correction methods.

**Convergence analysis.** We now state our convergence theorem for a convex, $\beta$-smooth function.

**Theorem 1.** *Let $f$ be a convex, $\beta$-smooth function with bounded gradients, then the iterates in Eq. (10) with $\eta = \frac{1}{\beta}$ converges at a rate of $O(\frac{1}{t})$.*

*Proof.* We largely follow the proof of (Bubeck et al., 2015) while adopting delayed gradients. As part of the proof, the momentum coefficient is set to $\gamma_t = \frac{t-2}{t}$ and we show that $\|\Delta_t\| = O(\frac{1}{t})$ using the bounded gradient assumption. Detailed proof can be found in Appendix A.3. $\square$

We do not claim the sublinear rate we derived is tight, and the rate or the constants in the bound may be improved. We leave any such analysis for future work. To our knowledge, this is the first time a variant of the Nesterov method is shown to converge in the presence of delayed gradients.

---

[2]Note, at time $t - \tau$, the weights $\mathbf{w}_{t-\tau}$ are extrapolated by $\mathbf{d}_{t-\tau}$ and the gradients are computed at the point $\mathbf{w}_{t-\tau} + \mathbf{d}_{t-\tau}$, to compensate for the "future" delay $\Delta_t$.

**Implementation details.** For transformer architectures, AdamW optimizer (Loshchilov, 2017) is known to be superior over SGD. To this end, we adopt the NAdam optimizer (Dozat, 2016) that incorporates the Nesterov method in Adam (Kingma, 2014) with decoupled weight decay for our language model training. Interestingly, both Adam and NAdam discount the gradient term by $(1 - \gamma_t)$, as they treat momentum as an exponential moving average of gradients. Even though the motivation of Adam is different, in this paper, we theoretically and empirically show the effectiveness of this discount factor for asynchronous optimization.

In the PyTorch implementation of NAdam (PyTorch Contributors, 2025), the momentum coefficient is warmed up to the $\beta_1$ value passed to the algorithm, which satisfies our assumption in Proposition 1 when $\beta_1$ is set close to 1. To this end, we use the *NAdam optimizer as is*, and use a large value for $\beta_1$ (0.99 in our experiments). It is remarkable that, with almost no modifications to an existing implementation, we show significant improvement over existing asynchronous optimization methods and provide theoretical justification. In fact, as we show in our experiments, simply changing the optimizer to NAdam improves other delay correction methods such as the learning rate discounting approach (Yang et al., 2021) and gradient forecasting methods (Zheng et al., 2017). However, such delay correction methods deteriorate the performance of NAdam as is, validating our insight that correcting delays in the weight space is more effective.

Since we build our method using the PipeDream framework (refer Sec. 2.2), at each stage we store a copy of weights for each active microbatch to ensure correct gradient computation. This amounts to storing $\tau_i$ copies of weights in stage $i$, therefore, the memory requirement of our method grows linearly with the delay (or number of stages). Even though, in practice, these weight stashes can be offloaded to the CPU effectively masking the memory requirement, for completeness, we also discuss a *no-weight-stash version* of our method below. Despite backpropagating through different sets of weights compared to the forward pass, surprisingly, this method is competitive to the synchronous method *without any additional memory requirements*.

## 3.2. Memory Efficient Version

As noted before, without weight stashing, the backpropagation is incorrect, *i.e.*, Eq. (3) takes the following form:

$$\tilde{\nabla} f_i(\mathbf{w}_i^{t-\tau_i}) = h_i(\mathbf{w}_i^t, \tilde{\mathbf{e}}_i^{t-\tau_{i+1}}), \tag{12}$$
$$\tilde{\mathbf{e}}_{i-1}^{t-\tau_i} = g_i(\mathbf{w}_i^t, \tilde{\mathbf{e}}_i^{t-\tau_{i+1}}).$$

Here, since $\mathbf{w}_i^t$ is used to backpropagate (instead of $\mathbf{w}_i^{t-\tau_i}$), the gradients are altered for the following stages. Hence, we denote the delayed error signal by $\tilde{\mathbf{e}}$ and the weight gradients by $\tilde{\nabla} f_i$ to indicate that they are altered. To compensate for this, we make two modifications: 1) stage-dependent learn-

ing rate, and 2) stage-dependent momentum coefficient.

Here, the idea is that earlier stages incur larger delays and also larger error accumulation due to incorrect backpropagation. Even if our variant of NAG can alleviate the issue with larger delays, larger error accumulation is still a problem. To this end, we further decrease the learning rate for earlier stages following the idea of (Yang et al., 2021). Additionally, the momentum coefficient is linearly increased from 0.9 to 0.99 from the last stage to the first one. Precisely, the learning rate $\eta$ and the momentum coefficient $\gamma$ for stage $i \in \{1, 2, \ldots, P\}$ are set as follows:

$$\eta_i^t = \frac{\eta}{\tau_i^{\rho_t}} \qquad \text{where } \rho_t = 1 - \min\left(\frac{t}{T}, 1\right), \qquad (13)$$

$$\gamma_i = 0.9 + \frac{P - i}{P} * 0.09 \,,$$

where $P$ is the number of stages. Note, the learning rate correction is only applied for the first $T$ iterations to stabilize training as in (Yang et al., 2021).

## 4. Related Work

**Asynchronous data parallel methods.** Data Parallelism (DP) is a traditional distributed training setting, where each device optimizes the full model and periodically synchronizes the model parameters. Asynchronous DP methods are well-studied within the theoretical framework and many gradient delay correction mechanisms have been developed (Agarwal & Duchi, 2011; Stich & Karimireddy, 2019; Assran et al., 2020). Notable methods that improve over the simple asynchronous SGD (Recht et al., 2011) include delay dependent learning rate (Barkai et al., 2019; Mishchenko et al., 2022), gradient forecasting with second-order information (Zheng et al., 2017), and look-ahead in the weight space (Hakimi et al., 2019). Apart from this, training dynamics of asynchronous DP methods have also been analyzed (Mitliagkas et al., 2016; Liu et al., 2024b) and some of these observations may be useful in the PP setting as well.

**Pipeline parallel methods.** The main objective of PP methods (Guan et al., 2024) is to improve pipeline utilization which led to the development of many pipeline scheduling strategies including GPipe (Huang et al., 2019), 1F1B (Narayanan et al., 2021b), and Zero Bubble (Qi et al., 2023). However, these methods suffer from synchronization bottlenecks. Asynchronous methods alleviate this bottleneck to achieve 100% pipeline utilization at the cost of gradient staleness. Notable gradient delay correction mechanisms for PP include weight stashing (Narayanan et al., 2019; 2021a), learning rate discounting (Yang et al., 2021), and direct weight prediction (Chen et al., 2018; Guan et al., 2019). Moreover, gradient forecasting methods developed for DP (Zheng et al., 2017) can also be employed. However, existing asynchronous PP methods are mainly empirical

and tested on small-scale image classification or language translation tasks. In contrast, for the first time, we demonstrate that asynchronous methods can surpass synchronous methods in 1B parameter scale language modelling tasks, in addition to providing theoretical convergence guarantees.

## 5. Experiments

### 5.1. Experimental Setup

We evaluate our method on the language modelling task using decoder-only architectures. We use three large-scale datasets: WikiText (WT) (Merity et al., 2016), BookCorpus (BC) (Zhu et al., 2015), and OpenWebText (OWT) (Gokaslan et al., 2019) datasets. For WikiText, we utilize the predefined training and validation splits, for the other datasets, we randomly select 10% of the training set as the held-out validation set. Our model architecture is based on NanoGPT (Karpathy, 2022) with no dropout. The base configuration includes a context length of 512, an embedding dimension of 768, 12 attention heads, and 8 layers, with approximately 134M parameters, and each layer is treated as a stage in our PP framework. We use the GP2 tokenizer (Radford et al., 2019) and train the model from scratch. Across all experiments, we maintain a microbatch size of 8, a learning rate $\eta$ of $3e$-4, and a weight decay of 0.01, unless otherwise specified. The update interval $K = 1$ for the asynchronous methods. The learning rate is obtained by tuning the performance of GPipe on the Wiki-Text dataset. All baseline methods employ the AdamW optimizer (Loshchilov, 2017). Each experiment is run for 50k iterations, with a linear warmup of 3k iterations starting from a learning rate of $1e$-7. Then, it is decayed to $3e$-5 following a cosine decay schedule.

We evaluate two existing asynchronous PP methods, PipeDream (Narayanan et al., 2019) and PipeMare (Yang et al., 2021), together with the synchronous GPipe (Huang et al., 2019) method. Unlike PipeDream, which stashes old weights, PipeMare estimates these weights using the velocity of weight updates. Moreover, PipeMare incorporates a learning rate discounting mechanism as described in Eq. (13), with $T$ set to 6k iterations. We implement PipeMare within the PipeDream framework. For GPipe, we use the `torch.distributed.pipelining` package, setting the number of microbatches to 4, which is limited by the GPU memory.[3] Due to differences in the underlying implementations, the absolute wall-clock time is not comparable between methods. We instead compare performance based on training iterations, or equivalently the amount of data processed.

---

[3]In our experiments, GPipe accumulates 4 microbatches for each weight update to reduce pipeline bubbles (Huang et al., 2019), and therefore it performs $4\times$ less updates for the *same amount of data* than asynchronous methods.

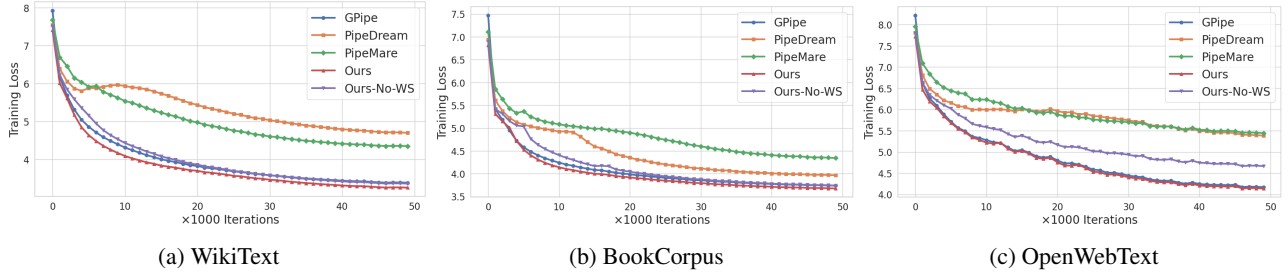

|       | (a) WikiText | (b) BookCorpus | (c) OpenWebText |
|-------|-------------|----------------|-----------------|

Figure 2: *Training trajectory comparison on three language modelling datasets. In all scenarios, our method significantly outperforms the asynchronous methods while surpassing the synchronous GPipe method throughout training. Our memory efficient version clearly outperforms the asynchronous methods while being competitive to GPipe in two out of three datasets.*

| Method | WT | BC | OWT | Memory |
|--------|-----|-----|------|--------|
| GPipe | 30.63 | 42.39 | 65.17 | $O(N)$ |
| PipeDream | 99.48 | 52.98 | 224.30 | $O(PN)$ |
| PipeMare | 71.38 | 76.93 | 239.13 | $O(N)$ |
| Ours | **27.72** | **39.85** | **62.86** | $O(PN)$ |
| Ours-No-WS | 29.90 | 42.61 | 108.20 | $O(N)$ |

Table 1: *Perplexity scores on the validation set at 50k iterations. The memory requirement of each method is shown in the last column, where $N$ is the number of parameters and $P$ is the number of pipeline stages. Ours outperforms all methods, including the synchronous GPipe method. Our memory efficient version (Ours-No-WS) is on par with GPipe on perplexity, while outperforming other asynchronous methods.*

Our proposed method is denoted as **Ours**, which employs the Nadam optimizer (Dozat, 2016) with decoupled weight decay and a momentum coefficient $\beta_1$ of 0.99. The memory-efficient variant of our method, **Ours-No-WS**, incorporates the same learning rate discounting as PipeMare (see Eq. (13)), with $T$ also set to 6k iterations. Unless otherwise specified, all experiments use the base architecture described above and are performed on the WikiText dataset with the aforementioned hyperparameters. All experiments are performed on a system equipped with 8 A10G GPUs.

### 5.2. Main Results

We analyze the training trajectories of our method against the baselines for the base architecture with 8 stages, as illustrated in Fig. 2. The results demonstrate that our method significantly outperforms existing asynchronous approaches and even surpasses GPipe across all three datasets. Our memory-efficient variant is competitive with GPipe, matching its performance on two out of the three datasets. Notably, the training trajectories of PipeDream and PipeMare reveal the optimization challenges inherent in asynchronous setups,[4] while our Nesterov-based delay correction effectively bridges the gap between asynchronous and synchronous

---

[4]To the best of our knowledge, this is the first time PipeDream and PipeMare are tested on large-scale language modelling tasks.

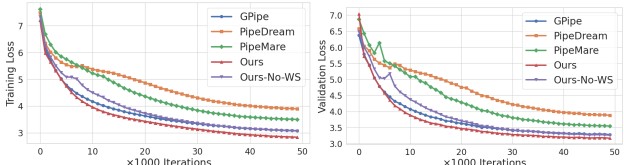

Figure 3: *Training and validation trajectories for the 1B parameter model. Similar to the base model, our method outperforms GPipe while the memory efficient version is competitive with GPipe.*

methods. Unlike PipeMare, which estimates old weights to facilitate correct backpropagation, our no-weight-stash version achieves superior performance without such estimation, relying instead on the Nesterov-based delay correction.

Although we primarily report training loss, the trends are consistent with validation loss as well. Validation loss comparison is provided in Fig. 9 in the appendix. For completeness, we include the perplexity scores on the validation set for all methods at 50k iterations in Table 1 along with the memory requirement. In this, we only consider the memory requirement for storing weights and do not consider activation memory as it is the same for all the methods.

### 5.3. Increasing the Model Size

To show the scalability of our approach, we train a **1B parameter model** in the asynchronous setting. We maintain the number of stages at 8 but increase the context length to 1024 and the embedding dimension to 2688, with 24 attention heads. A learning rate of $1e$-4 is used for all methods. These experiments are performed on a system equipped with 8 A100 GPUs.

As shown in Fig. 3, the results are consistent with those of the base model. Specifically, our method significantly outperforms all baselines, including GPipe, throughout training. Notably, our no-weight-stash variant matches the performance of GPipe. Aside from learning rate adjustment, no changes were made to the method or its hyperparameters. This large-scale experiment demonstrates the merits of our method and the practicality of asynchronous PP optimization for language model training.

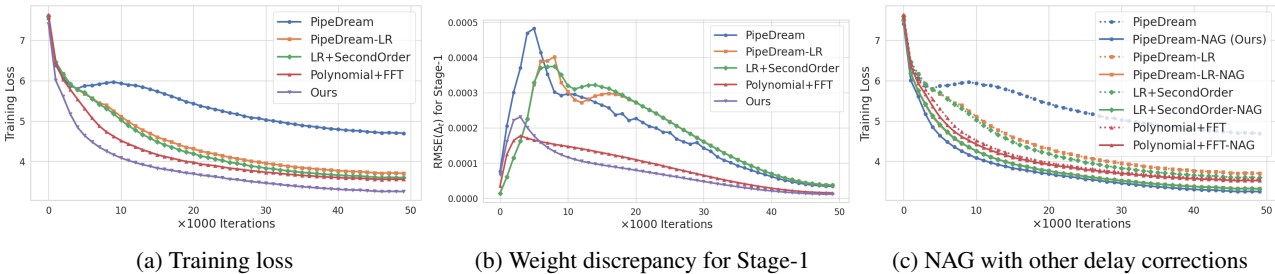

|     |     |     |
| --- | --- | --- |
| (a) Training loss | (b) Weight discrepancy for Stage-1 | (c) NAG with other delay corrections |

Figure 4: *Comparison with other delay correction methods on WikiText. Our method, outperforms all other delay correction methods in terms of training loss and weight discrepancy. Additionally, NAG improves all previous delay correction methods, while NAG alone yields the best performance.*

### 5.4. Other Delay Correction Methods

We compare our method with the following delay correction mechanisms which were originally developed for asynchronous optimization in the DP setting.

**PipeDream-LR.** The learning rate discounting method (Yang et al., 2021; Mishchenko et al., 2022) which employs a delay dependent discounting of the learning rate as noted in Eq. (13) where $T$ is set to 6k.

**LR-SecondOrder.** On top of the learning rate correction above, we forecast the gradients to the current step using the second-order Taylor expansion of the loss as in (Zheng et al., 2017). The implementation is similar to (Zheng et al., 2017) where the diagonal of the Fisher matrix is used to approximate the Hessian.

**Polynomial+FFT.** In this approach, we frame gradient forecasting as a time series prediction problem, leveraging historical gradient data to predict future gradients. Specifically, we employ a second-order polynomial to model trends and utilize Fast Fourier Transform (FFT) to capture any periodic signals. The history size is set to 8. This is a well-known method in time series forecasting literature (Bloomfield, 2004), which we adopt for gradient delay correction.

In addition to monitoring training loss, we also evaluate the Root-Mean-Square Error (RMSE) of the weight discrepancy $\Delta_t$ (as in Eq. (9)) at the first stage, which experiences the largest delay. This metric is named gap in (Hakimi et al., 2019), which directly measures the effectiveness of delay correction, where smaller gap indicates better delay correction. The results are presented in Fig. 4.

Among the above delay correction methods, polynomial fitting is the most effective. However, our simple Nesterov-based approach outperforms all other sophisticated techniques in both training loss and weight discrepancy. Notably, our method is complementary to those strategies and further enhances their performance. Although combining Nesterov with other delay correction mechanisms diminishes its effectiveness, supporting our hypothesis that correcting delays in the weight space is more impactful than other approaches.

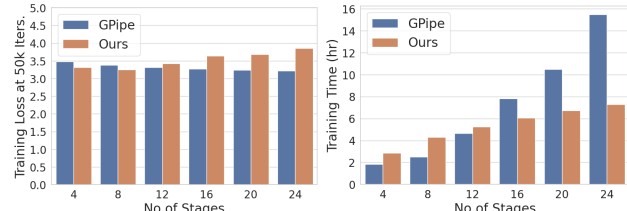

Figure 5: *Performance with respect to the number of stages. Even though, performance slightly degrades for our method compared to GPipe, the training time increase is exponentially larger for GPipe.*

### 5.5. Increasing the Number of Stages

To evaluate scalability with respect to the number of stages, we increase the number of layers in the base model whilst maintaining the same embedding dimension. We compare the results against GPipe in terms of training loss and the percentage increase in training time. For configurations with 20 and 24 stages, the learning rate is reduced to $1e\text{-}4$ for our method, while all other hyperparameters remain unchanged. The results are presented in Fig. 5.

Since the delay grows with the number of stages, the performance of our method degrades slightly as expected. However, due to 100% pipeline utilization, the percentage increase in runtime for our approach is significantly lower compared to GPipe. Concretely, for GPipe, 24-stage model takes $8.5\times$ more time compared to the 4-stage model, however, our model is only $2.5\times$ slower. This highlights the trade-off between performance and runtime efficiency, underscoring the advantages of asynchronous optimization. This runtime discrepancy is more pronounced when a faster GPU is used, such as A100, as observed in our 1B model experiments (refer to Fig. 10 in the appendix).

### 5.6. Ablation Study

To understand the effect of momentum coefficient, we vary the momentum coefficient and report the performance of our base model on the WikiText dataset. In addition to training loss, we also measure the cosine similarity be-

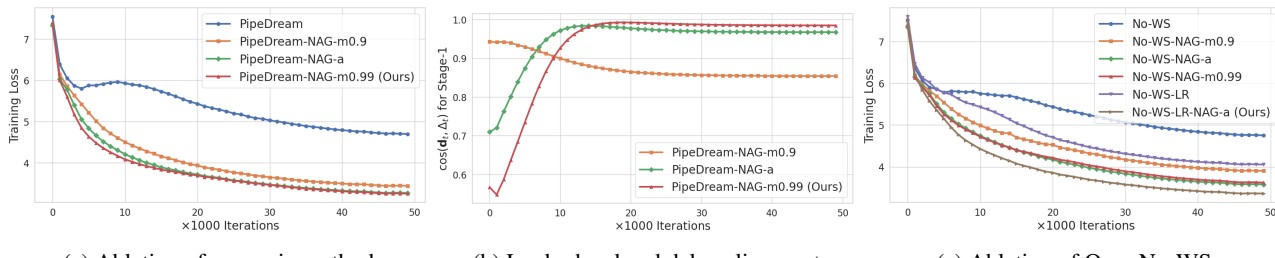

(a) Ablation of our main method     (b) Look-ahead and delay alignment     (c) Ablation of Ours-No-WS

Figure 6: *Ablation study of our methods. Constant momentum coefficient of 0.99 performs slightly better than the adaptive version (denoted with '-a') and it also shows the best alignment in (b). For the memory efficient version, in addition to adaptive momentum, delay dependent learning rate discounting also helps as shown in (c).*

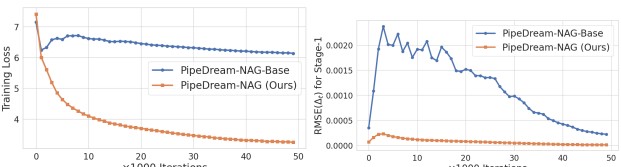

Figure 7: *Our approach with and without the gradient discounting term for the Nesterov method. Without the discounting term, training is significantly disrupted due to gradient staleness, validating our insight.*

tween the look-ahead direction $\mathbf{d}_t$ and the weight difference $\Delta_t = \mathbf{w}_t - \mathbf{w}_{t-\tau}$ at the first stage, where the discrepancy is most pronounced. The results are presented in Fig. 6.

As expected, increasing the momentum coefficient from 0.9 to 0.99 leads to improved performance. However, using an adaptive momentum coefficient, as defined in Eq. (13), results in slightly worse performance compared to a fixed value of 0.99 for our main method. The influence of the momentum coefficient on the alignment between the look-ahead step and the weight difference is also empirically demonstrated, matching our theoretical insight. On the other hand, for the memory-efficient version, a delay-adaptive momentum coefficient performs better, and incorporating learning rate discounting further enhances performance.

**Effect of gradient discounting.** As outlined in Sec. 3.1, the standard implementation of NAdam incorporates a discount factor of $(1-\gamma_t)$ for the gradient term. To demonstrate the necessity of this discounting factor for our approach, we train a model using a modified optimizer where this term is removed, denoted as PipeDream-NAG-Base. The results are presented in Fig. 7.

Without this discount factor, the method struggles due to gradient staleness and fails to achieve a training loss comparable to our method. Notably, the weight discrepancy at the first stage is an order of magnitude larger than that observed with our gradient discounting approach. Empirically, this strongly validates the significance of the discounting factor.

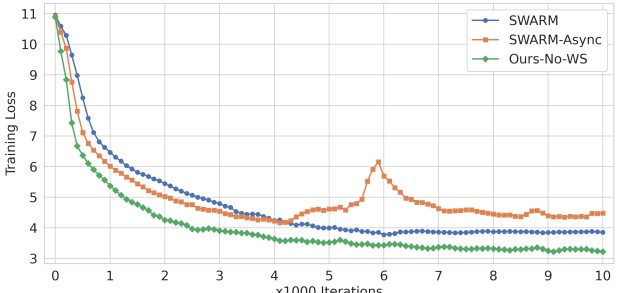

Figure 8: *Training trajectory comparison in SWARM. Our method significantly outperforms both the synchronous and asynchronous versions of SWARM.*

### 5.7. Realistic Decentralized Training

Finally, to stress test our method, we evaluate our approach in a realistic decentralized setting using SWARM (Ryabinin et al., 2023). SWARM is built using the Hivemind framework (Ryabinin & Gusev, 2020; Ryabinin et al., 2020) and it supports fault-tolerant, pipeline-parallel training with multiple worker nodes per stage (*i.e.*, DP at each stage) connected over the internet. Natively, it employs gradient accumulation akin to synchronous training and the workers at each stage are synchronized periodically. We refer the interested reader to Ryabinin et al. (2023) for more details.

Our architecture and the hyperparameters are similar to the base model experiments but adapted to accommodate SWARM and we train on the WikiText dataset. More details can be found in Appendix B.1. We compare three variants: 1) the standard, synchronous setting (SWARM); 2) an asynchronous setting with local updates for every microbatch and a periodic stage-wise weight synchronization (SWARM-Async); and 3) SWARM-Async with our *no-weight-stash* version (Ours-No-WS). Note that weight stashing is not applicable in SWARM. Results are reported in Fig. 8.

Our method clearly outperforms both the synchronous and asynchronous versions of SWARM throughout training. Notably, SWARM-Async shows training instability even with a lower learning rate (refer to Appendix B.1), whereas our method shows stable convergence. Validation loss also follows a similar trend as shown in Fig. 13 in the appendix.

# 6. Conclusion

We introduce a novel variant of NAG to alleviate gradient staleness in asynchronous PP optimization. Theoretically, we show that our algorithm converges at a sublinear rate for convex, smooth functions in the non-stochastic setting with fixed delay in gradients. Practically, adopting our method is as simple as switching the optimizer and changing the value of an existing hyperparameter. To show the merits of our approach, we performed large-scale experiments on language modelling tasks using models up to 1B parameters. In all our experiments, our method consistently outperformed previous asynchronous methods as well as the synchronous GPipe method. The behaviour is consistent even in decentralized experiments in SWARM. In the future, we intend to investigate the tightness of our convergence rate and extend our approach to PP with stage-wise DP setting, in a heterogenous decentralized training environment.

## Impact Statement

Our method establishes the feasibility of asynchronous PP optimization for billion-scale language model training, improving efficiency in distributed and decentralized settings. By enabling more effective parallelization, it has the potential to reduce energy consumption and training costs while increasing accessibility to large-scale model training via decentralized infrastructures. As a theoretical contribution, its societal impact is application-dependent, with no direct consequences attributable to the method itself.

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

## A. Theoretical Analysis

We first restate our NAG updates, and then turn to the proofs.

### A.1. Nesterov Method for Delayed Gradients

Our variant of NAG performs the following iterations starting from an initial point $\mathbf{w}_1$:

$$\mathbf{d}_t = \gamma_t(\mathbf{w}_t - \mathbf{w}_{t-1}) \,, \tag{14}$$
$$\mathbf{w}_{t+1} = \mathbf{w}_t + \mathbf{d}_t - \eta(1 - \gamma_t)\nabla f(\bar{\mathbf{w}}_t + \bar{\mathbf{d}}_t) \,.$$

Here, $\gamma_1 = 0$ and $\bar{\mathbf{w}}_t$ and $\bar{\mathbf{d}}_t$ denote the delayed versions of weights and look-ahead, respectively, *i.e.*,

$$\bar{\mathbf{w}}_t = \mathbf{w}_{t-\tau} = \mathbf{w}_t - \Delta_t \,, \tag{15}$$
$$\bar{\mathbf{d}}_t = \mathbf{d}_{t-\tau} \,,$$

where $\tau \geq 0$ is the delay, and for simplicity we assume it to be fixed. The main difference to the widely used NAG version is the $(1 - \gamma_t)$ term for the gradients.

Let us introduce some notations that might be helpful later:

$$\bar{\mathbf{g}}_t = \nabla f(\bar{\mathbf{w}}_t + \bar{\mathbf{d}}_t) \,, \tag{16}$$
$$\bar{\mathbf{h}}_t = -\eta(1 - \gamma_t)\bar{\mathbf{g}}_t \,,$$
$$\bar{\Delta}_t = \mathbf{w}_t + \mathbf{d}_t - \left(\bar{\mathbf{w}}_t + \bar{\mathbf{d}}_t\right) = \Delta_t + \mathbf{d}_t - \bar{\mathbf{d}}_t \,.$$

$\mathbf{g}_t$ and $\mathbf{h}_t$ are analogously defined. From above, we may note the following identities:

$$\mathbf{w}_{t+1} = \mathbf{w}_t + \mathbf{d}_t + \bar{\mathbf{h}}_t \,, \tag{17}$$
$$\mathbf{d}_{t+1} = \gamma_{t+1}(\mathbf{d}_t + \bar{\mathbf{h}}_t) \,,$$

We are now ready to prove that the look-ahead step acts as delay correction when the momentum coefficient $\gamma_t$ is chosen appropriately. Then, we prove the convergence rate.

### A.2. Look-ahead as Delay Correction

**Proposition 2.** Let $\gamma_t$ be an increasing sequence such that $\lim_{t\to\infty} \gamma_t = 1$ then, $\lim_{t\to\infty} \cos(\Delta_t, \bar{\mathbf{d}}_t) = 1$, where $\cos(\cdot, \cdot)$ is the cosine similarity.

*Proof.* Let us expand the delay term:

$$\Delta_t = \mathbf{w}_t - \mathbf{w}_{t-\tau} = \sum_{i=t-\tau+1}^{t} \mathbf{w}_i - \mathbf{w}_{i-1} = \sum_{i=t-\tau+1}^{t} \frac{\mathbf{d}_i}{\gamma_i} = \sum_{i=t-\tau+1}^{t} \mathbf{d}_{i-1} + \bar{\mathbf{h}}_{i-1} = \sum_{i=1}^{\tau} \mathbf{d}_{t-i} + \bar{\mathbf{h}}_{t-i} \,. \tag{18}$$

Note, all $\mathbf{d}_{t-i} + \bar{\mathbf{h}}_{t-i}$ can be written in terms of $\bar{\mathbf{d}}_t = \mathbf{d}_{t-\tau}$:

$$\mathbf{d}_{t-i} + \bar{\mathbf{h}}_{t-i} = \Pi_{j=t-\tau+1}^{t-i}\gamma_j \, \mathbf{d}_{t-\tau} + \sum_{k=t-\tau}^{t-i} \Pi_{j=k+1}^{t-i}\gamma_j \, \bar{\mathbf{h}}_k \,, \tag{19}$$

$$= \left(\Pi_{j=t-\tau+1}^{t-i}\gamma_j\right) \mathbf{d}_{t-\tau} - \eta \sum_{k=t-\tau}^{t-i} \left(\Pi_{j=k+1}^{t-i}\gamma_j\right) (1 - \gamma_k) \, \bar{\mathbf{g}}_k \,.$$

Now we can write $\Delta_t$ in terms of $\bar{\mathbf{d}}_t$ as follows:

$$\Delta_t = \sum_{i=1}^{\tau} \left[ \left(\Pi_{j=t-\tau+1}^{t-i}\gamma_j\right) \bar{\mathbf{d}}_t - \eta \sum_{k=t-\tau}^{t-i} \left(\Pi_{j=k+1}^{t-i}\gamma_j\right) (1 - \gamma_k) \, \bar{\mathbf{g}}_k \right] \,. \tag{20}$$

When, $\gamma_t \to 1$, the gradient term vanishes, and $\cos(\Delta_t, \bar{\mathbf{d}}_t) \to 1$. $\qquad \square$

Similarly, by writing $\Delta_t$ in terms of $\mathbf{d}_t$, one can show that $\cos(\Delta_t, \mathbf{d}_t) \to 1$. This effectively shows that when the momentum coefficient $\gamma_t \to 1$, the optimization trajectory becomes smooth and the delay and the look-ahead directions align.

### A.3. Convergence Analysis

In this section, we will prove that NAG with delayed gradients converges at a rate of $O(\frac{1}{t})$ similar to the standard gradient descent. The sequence $\gamma_t$ will be set as part of the convergence proof. Even though, due to the delay the faster $O(\frac{1}{t^2})$ rate is not achieved, in practice NAG is as effective as the synchronous alternatives despite the staleness in gradients.

**Theorem 2.** *Let $f$ be a convex, $\beta$-smooth function with bounded gradients, then the iterates in Eq. (14) with $\eta = \frac{1}{\beta}$ converges at a rate of $O(\frac{1}{t})$.*

*Proof.* Our proof largely follows the proof of (Bubeck et al., 2015) while adopting the delayed gradients. Let us first introduce a few useful inequalities:

$$f(\mathbf{x}) - f(\mathbf{y}) - \nabla f(\mathbf{y})^T(\mathbf{x} - \mathbf{y}) \le \frac{\beta}{2}\|\mathbf{x} - \mathbf{y}\|^2 , \qquad f \text{ is } \beta\text{-smooth} \qquad (21)$$

$$f(\mathbf{x}) - f(\mathbf{y}) \le \nabla f(\mathbf{x})^T(\mathbf{x} - \mathbf{y}) , \qquad \text{convex}$$

$$\|\nabla f(\mathbf{x}) - \nabla f(\mathbf{y})\| \le \beta\|\mathbf{x} - \mathbf{y}\| . \qquad \beta\text{-smooth}$$

Applying the first inequality above to the update equation, while using Eq. (16) and $\eta = \frac{1}{\beta}$:

$$f(\mathbf{w}_{t+1}) - f(\bar{\mathbf{w}}_t + \bar{\mathbf{d}}_t) \le \bar{\mathbf{g}}_t \cdot \left(\bar{\Delta}_t - \frac{1 - \gamma_t}{\beta}\bar{\mathbf{g}}_t\right) + \frac{\beta}{2}\left\|\bar{\Delta}_t - \frac{1 - \gamma_t}{\beta}\bar{\mathbf{g}}_t\right\|^2 , \qquad (22)$$

$$= \bar{g}_t \cdot \bar{\Delta}_t - \frac{1 - \gamma_t}{\beta}\|\bar{\mathbf{g}}_t\|^2 + \frac{\beta}{2}\left(\|\bar{\Delta}_t\|^2 - \frac{2(1 - \gamma_t)}{\beta}\bar{\mathbf{g}}_t \cdot \bar{\Delta}_t + \frac{(1 - \gamma_t)^2}{\beta^2}\|\bar{\mathbf{g}}_t\|^2\right) ,$$

$$= \frac{\beta}{2}\|\bar{\Delta}_t\|^2 + \gamma_t\bar{\mathbf{g}}_t \cdot \bar{\Delta}_t + \frac{\gamma_t^2 - 1}{2\beta}\|\bar{\mathbf{g}}_t\|^2 .$$

Here, $\cdot$ denotes the dot product. Similarly, using convexity:

$$f(\bar{\mathbf{w}}_t + \bar{\mathbf{d}}_t) - f(\mathbf{w}_t) \le \bar{\mathbf{g}}_t \cdot (\bar{\mathbf{d}}_t - \Delta_t) = \bar{\mathbf{g}}_t \cdot (\mathbf{d}_t - \bar{\Delta}_t) . \qquad (23)$$

Now summing Eq. (22) and Eq. (23):

$$\delta_{t+1} - \delta_t = f(\mathbf{w}_{t+1}) - f(\mathbf{w}_t) \le \frac{\beta}{2}\|\bar{\Delta}_t\|^2 + (\gamma_t - 1)\bar{\mathbf{g}}_t \cdot \bar{\Delta}_t + \frac{\gamma_t^2 - 1}{2\beta}\|\bar{\mathbf{g}}_t\|^2 + \bar{\mathbf{g}}_t \cdot \mathbf{d}_t . \qquad (24)$$

where $\delta_{t+1} = f(\mathbf{w}_{t+1}) - f(\mathbf{w}^*)$. Similarly,

$$\delta_{t+1} = f(\mathbf{w}_{t+1}) - f(\mathbf{w}^*) \le \frac{\beta}{2}\|\bar{\Delta}_t\|^2 + (\gamma_t - 1)\bar{\mathbf{g}}_t \cdot \bar{\Delta}_t + \frac{\gamma_t^2 - 1}{2\beta}\|\bar{\mathbf{g}}_t\|^2 + \bar{\mathbf{g}}_t \cdot (\mathbf{w}_t + \mathbf{d}_t - \mathbf{w}^*) . \qquad (25)$$

Now, suppose $\lambda_t > 1$ be a sequence. By multiplying Eq. (24) by $(\lambda_t - 1)$ and adding to Eq. (25):

$$\lambda_t\delta_{t+1} - (\lambda_t - 1)\delta_t \le \frac{\lambda_t\beta}{2}\|\bar{\Delta}_t\|^2 + \lambda_t(\gamma_t - 1)\bar{\mathbf{g}}_t \cdot \bar{\Delta}_t + \frac{\lambda_t(\gamma_t^2 - 1)}{2\beta}\|\bar{\mathbf{g}}_t\|^2 + \bar{\mathbf{g}}_t \cdot (\mathbf{w}_t + \lambda_t\mathbf{d}_t - \mathbf{w}^*) , \qquad (26)$$

$$= \Omega_t - \frac{\lambda_t(1 + \gamma_t)\beta}{2(1 - \gamma_t)}\bar{\mathbf{h}}_t^2 - \frac{\beta}{1 - \gamma_t}\bar{\mathbf{h}}_t \cdot (\mathbf{w}_t + \lambda_t\mathbf{d}_t - \mathbf{w}^*) , \qquad \mathbf{A}$$

$$\le \Omega_t - \frac{\lambda_t\beta}{2(1 - \gamma_t)}\bar{\mathbf{h}}_t^2 - \frac{\beta}{1 - \gamma_t}\bar{\mathbf{h}}_t \cdot (\mathbf{w}_t + \lambda_t\mathbf{d}_t - \mathbf{w}^*) , \qquad \mathbf{B}$$

$$= \Omega_t - \frac{\beta}{2\lambda_t(1 - \gamma_t)}\left(\|\lambda_t\bar{\mathbf{h}}_t + \mathbf{w}_t + \lambda_t\mathbf{d}_t - \mathbf{w}^*\|^2 - \|\mathbf{w}_t + \lambda_t\mathbf{d}_t - \mathbf{w}^*\|^2\right) , \qquad \mathbf{C}$$

where, **A** substitutes $\Omega_t = \frac{\lambda_t\beta}{2}\|\bar{\Delta}_t\|^2 + \lambda_t(\gamma_t - 1)\bar{\mathbf{g}}_t \cdot \bar{\Delta}_t$, **B** is due to $0 < \gamma_t < 0, \lambda_t > 0, \beta > 0$, and **C** follows from $a^2 + 2ab = (a + b)^2 - b^2$.

Now, to enable telescoping sum, we want,

$$\lambda_t\bar{\mathbf{h}}_t + \mathbf{w}_t + \lambda_t\mathbf{d}_t - \mathbf{w}^* = \mathbf{w}_{t+1} + \lambda_{t+1}\mathbf{d}_{t+1} - \mathbf{w}^* = \mathbf{w}_t + \mathbf{d}_t + \bar{\mathbf{h}}_t + \lambda_{t+1}\gamma_{t+1}(\mathbf{d}_t + \bar{\mathbf{h}}_t) - \mathbf{w}^* . \qquad (27)$$

To this end, we set,

$$1 + \lambda_{t+1}\gamma_{t+1} = \lambda_t . \tag{28}$$

Now, let the sequence $\lambda_t = t$, then,

$$\gamma_t = \frac{t-2}{t} , \qquad \text{and} \qquad 1 - \gamma_t = \frac{2}{t} . \tag{29}$$

Now, let $\mathbf{u}_t = \frac{\beta}{2}\|\mathbf{w}_t + \lambda\mathbf{d}_t - \mathbf{w}^*\|^2$. Then,

$$\lambda_t\delta_{t+1} - (\lambda_t - 1)\delta_t \le \Omega_t + \frac{1}{\lambda_t(1 - \gamma_t)}(\mathbf{u}_t - \mathbf{u}_{t+1}) , \tag{30}$$

$$\lambda_t\delta_{t+1} - \lambda_{t-1}\delta_t \le \Omega_t + \frac{1}{2}(\mathbf{u}_t - \mathbf{u}_{t+1}) ,$$

$$\lambda_t\delta_{t+1} - \lambda_0\delta_1 \le \sum_{k=1}^{t}\Omega_k + \frac{1}{2}(\mathbf{u}_1 - \mathbf{u}_{t+1}) , \qquad\qquad \sum_{k=1}^{t} \text{ for both sides}$$

$$\lambda_t\delta_{t+1} \le \sum_{k=1}^{t}\Omega_k + \frac{1}{2}\mathbf{u}_1 , \qquad\qquad \lambda_0 = 0, \mathbf{u}_{t+1} \ge 0$$

$$\delta_{t+1} \le \frac{1}{t}\sum_{k=1}^{t}\Omega_k + \frac{\beta}{2t}\|\mathbf{w}_1 - \mathbf{w}^*\|^2 , \qquad\qquad \gamma_1 = 0$$

To have the rate $O(\frac{1}{t})$, it remains to show that $\sum_{k=1}^{t}\Omega_k$ grows much slower than $O(t)$. To this end, using the bounded gradients assumption we show $\|\mathbf{w}_{t+1} - \mathbf{w}_t\| = O(\frac{1}{t})$. We prove this by induction. The base case for $t \le \tau$ can be enforced using an appropriate warmup phase. Suppose $\|\mathbf{w}_t - \mathbf{w}_{t-1}\| = O(\frac{1}{t})$. Then,

$$\|\mathbf{w}_{t+1} - \mathbf{w}_t\|^2 = \|\mathbf{w}_t + \mathbf{d}_t + \bar{\mathbf{h}}_t - \mathbf{w}_t\|^2 , \tag{31}$$

$$= \|\mathbf{d}_t + \bar{\mathbf{h}}_t\|^2 ,$$

$$= \gamma_t^2\|\mathbf{w}_t - \mathbf{w}_{t-1}\|^2 + \frac{(1 - \gamma_t)^2}{\beta^2}\|\bar{\mathbf{g}}_t\|^2 - \frac{2\gamma_t(1 - \gamma_t)}{\beta}\bar{\mathbf{g}}_t \cdot (\mathbf{w}_t - \mathbf{w}_{t-1}) ,$$

$$\le \gamma_t^2\|\mathbf{w}_t - \mathbf{w}_{t-1}\|^2 + \frac{(1 - \gamma_t)^2}{\beta^2}\|\bar{\mathbf{g}}_t\|^2 + \frac{2\gamma_t(1 - \gamma_t)}{\beta}\|\bar{\mathbf{g}}_t\|\|\mathbf{w}_t - \mathbf{w}_{t-1}\| ,$$

$$= O\left(\frac{1}{t^2}\right) . \qquad \gamma_t, \|\bar{\mathbf{g}}_t\| = O(1), 1 - \gamma_t = O(\tfrac{1}{t}), \text{and}, \|\mathbf{w}_t - \mathbf{w}_{t-1}\| = O(\tfrac{1}{t})$$

Now, it is easy to see that $\|\mathbf{d}_t - \bar{\mathbf{d}}_t\| = O(\frac{1}{t})$ as $\gamma_t = O(1)$ and $\|\mathbf{w}_t - \mathbf{w}_{t-1}\| = O(\frac{1}{t})$. Consequently, $\|\bar{\Delta}_t\| = O(\frac{1}{t})$ due to bounded delay $\tau$. Consider,

$$\Omega_t = \frac{\lambda_t\beta}{2}\|\bar{\Delta}_t\|^2 + \lambda_t(\gamma_t - 1)\bar{\mathbf{g}}_t \cdot \bar{\Delta}_t , \tag{32}$$

$$\le \frac{t\beta}{2}\|\bar{\Delta}_t\|^2 + 2\|\bar{\mathbf{g}}_t\|\|\bar{\Delta}_t\| ,$$

$$= O\left(\frac{1}{t}\right) .$$

Therefore,

$$\sum_{k=1}^{t}\Omega_k = \sum_{k=1}^{t}O\left(\frac{1}{k}\right) = O(\ln t) , \tag{33}$$

which completes the proof. $\qquad\qquad\qquad\qquad\qquad\qquad\qquad\qquad\qquad\qquad\qquad\qquad\qquad\qquad\square$

Note here that $\bar{\Delta}_t$ depends on the delay $\tau$. Despite it being a constant, it may influence the convergence rate if it is sufficiently large. The proof largely follows the existing proof of NAG and analogously it may be extended to non-convex functions. We leave any such analysis to future work.

Furthermore, it may be intuitive to think of our discounting term $(1 - \gamma_t)$ as a learning rate discounting mechanism by considering $\eta_t = \frac{1-\gamma_t}{\beta} = O(\frac{1}{t})$. Analogously, relationships may be drawn from the convergence proof of such methods (Mishchenko et al., 2022). Nevertheless, in practical deep learning optimization, having a separate $\eta$ provides greater control over the learning rate schedules and our approach is consistently better than the learning rate discounting mechanism as shown in our experiments.

## B. Experiments

### B.1. SWARM Training Configuration

We use the SWARM baseline from Ryabinin et al. (2023) for our large-scale decentralized training framework. For all baselines, the model used is a Transformer language model with architecture similar to that in prior work (Brown et al., 2020; Wang & Komatsuzaki, 2021). Our SWARM configuration consists of 3 worker nodes per stage, for a total of 24 worker nodes. Each worker node is assigned an NVIDIA L4 GPU. We assign 24 trainer nodes to serve the entire pipeline, where each trainer node has a 4-core Intel Cascade Lake CPU with a base clock of 2.2 GHz and 32 GB of RAM.

We use the following layer configuration for all baselines: embedding dimension of 768 with 6 attention heads, the hidden layer dimension of the FFN layer of 3072 with 8 layers. Each layer is assigned to its own stage in the pipeline. The microbatch size used is 8 and sequence length is 2048. We employ a learning rate of $2e$-$4$ for the synchronous SWARM setting and our Nesterov-adapted approach. For the asynchronous variant of SWARM, we use a reduced learning rate of $5e$-$5$, due to training instability (and divergence) observed at higher learning rates.

A linear warmup was used for all baselines up to 1k steps and our Nesterov-adapted approach employs a stage-dependent learning rate up to 2k steps ($T$ in Eq. (13)) and the momentum coefficient $\beta_1$ is set as per Eq. (13). For the other methods, a default value of $\beta_1 = 0.9$ is used. All methods were trained for a total of 10k iterations, using a stage-wise all-reduce batch size of 256.

### B.2. Additional Results

We provide validation loss trajectories for the main experiments in Fig. 9, training loss vs time for the 1B model in Fig. 10, and more ablation results in Fig. 11. Furthermore, a comparison with the recent XPipe method (Guan et al., 2019) is provided in Fig. 12. Finally, validation loss trajectories for the SWARM experiments are provided in Fig. 13.

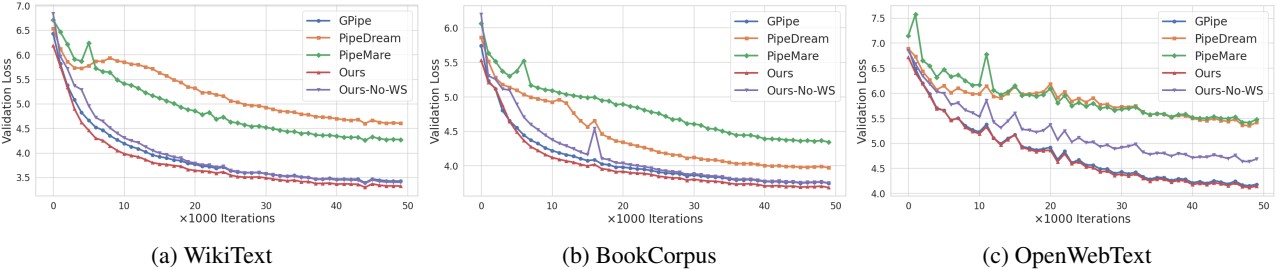

| (a) WikiText | (b) BookCorpus | (c) OpenWebText |

Figure 9: *Validation loss trajectory for the base model. The behaviour is the same as training loss where our method consistently outperforms all methods including GPipe.*

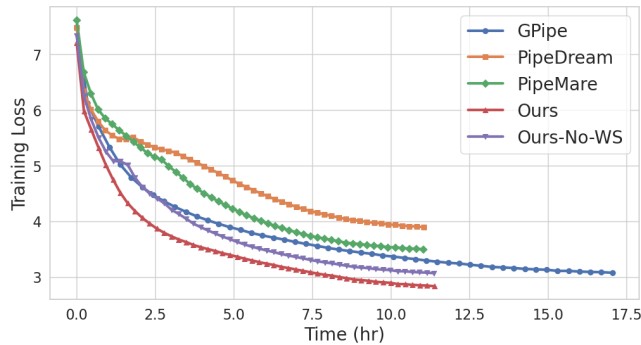

Figure 10: *Training loss* vs. *wall-clock time for the 1B model, where all models were trained for 50k iterations. For faster GPUs (A100 in this case), the runtime discrepancy between GPipe and other asynchronous methods is more pronounced. Note, overhead of our method over PipeDream is negligible.*

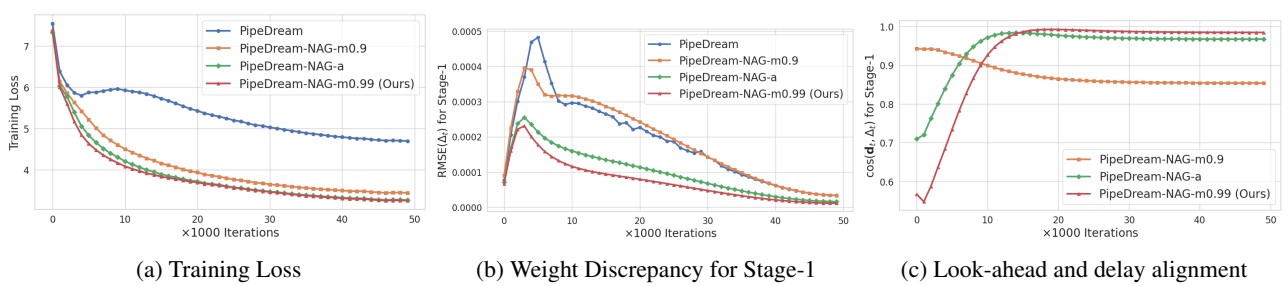

(a) Training Loss      (b) Weight Discrepancy for Stage-1      (c) Look-ahead and delay alignment

Figure 11: *Ablation study of our main methods additionally showing weight discrepancy at Stage-1 in (b). Constant momentum coefficient of 0.99 performs slightly better than the adaptive version (denoted with '-a'). It also shows the best delay correction and alignment between look-ahead and delay.*

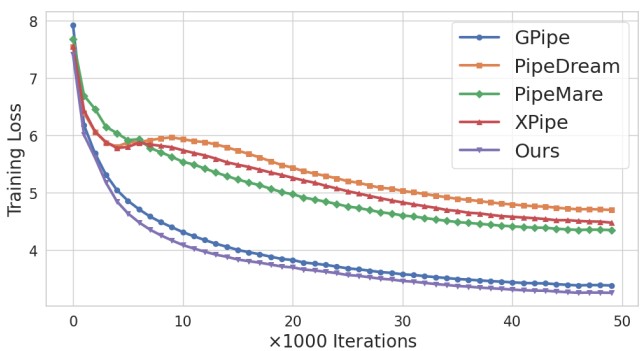

Figure 12: *WikiText results for the base model for the XPipe method (Guan et al., 2019) for completeness. XPipe is a direct weight prediction method, that extrapolates the previous AdamW step based on the delay. We implemented XPipe following the description from the paper, however, we were unable to reproduce its reported performance on our large-scale language model training. Note, only small-scale image classification experiments were reported in the paper and in our experiments all methods including PipeDream perform similarly (within $\sim 1$ pps) on those datasets.*

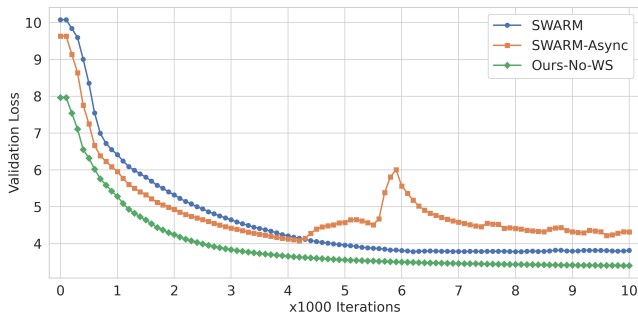

Figure 13: *Validation loss on WikiText for the SWARM experiment. The observed performance follows a similar trend to the training performance, where our method significantly outperforms the other methods.*

