# OpenReview forum: "Nesterov Method for Asynchronous Pipeline Parallel Optimization"
_ICML.cc/2025/Conference — ICML 2025 poster_

### Official Review · Reviewer_xheX · 2025-02-19

**Overall Recommendation:** 3

**Summary:**

In this paper, the authors proposed a Nesterov Accelerated Gradient algorithm for asynchronous pipeline parallel optimization. It use Nesterov acceleration to reduce the negative impact of delay in asynchronous iterations. The authors firstly provide the convergence guarantee that the Nesterov acceleration can help the asynchronous pipeline parallel algorithm achieve the $\mathcal{O}(1/T)$ convergence rate. Experiments on LLMs also validate the benefits of the proposed algorithms.

**Claims And Evidence:**

Yes.

**Essential References Not Discussed:**

NA

**Experimental Designs Or Analyses:**

The experiments have shown the benefits of the proposed algorithms. However, the reviewer suggests the authors present some results on the validation set so as to illustrate the value in practical application. Moreover, it is better for the authors to offer their core codes.

**Methods And Evaluation Criteria:**

Yes.

**Other Comments Or Suggestions:**

NA

**Other Strengths And Weaknesses:**

**Strengths**
1. Good presentation and writing.
2. Detailed analysis on the insight of algorithm design.

**Weaknesses**
1. There is a typo in Eq. (5). Please take a check and revision.
2. The convergence analysis is based on gradient descent without any stochastic noise. However, the convergence on stochastic algorithms like SGD or Adam may be essential for LLMs due to their small batch size.

**Questions For Authors:**

1. Whether the author can provide the experimental results on validation?
2. Whether the author can provide a theoretical analysis based on stochastic algorithms？

The reviewer may consider raising the score on overall recommendation according to the response to the weaknesses and questions.

**Relation To Broader Scientific Literature:**

NA

**Theoretical Claims:**

The reviewer has taken a check on the theoretical results in this paper and not found any fatal mistakes. However, one limitation of the theory is that the convergence analysis focus on deterministic scenarios. The theoretical claims will be better if the authors provide the analysis of the stochastic scenarios.

---

> ### Author Rebuttal · Authors · 2025-03-30
>
> We thank the reviewer for encouraging comments and address the specific points below.
>
> ## Results on the validation set
>
> We have **already provided the results on the validation set** in Table 1 (perplexity scores) and Fig. 3,9 (validation loss curves). Additionally, as requested by Reviewer tQWW, we have included some generated text as qualitative results in the response to tQWW.
>
> ## Theoretical analysis in the stochastic setting
>
> **We followed the standard practice in machine learning literature to provide theoretical analysis in a simplified setting and validate it using realistic large-scale experiments**. Many seminal works can be pointed out as examples (Kaifeng and Li 2019, Soudry et al 2018, Zhihui, et al. 2021). Furthermore, there have been instances where convergence analysis derived for non-stochastic settings are shown to be validated by experiments in the stochastic settings (Wu et al 2023, Arora et al 2018). We believe ours is one such example. Rigorous theoretical analysis matching the practical setting of large-scale language models requires significant research effort and we believe it would warrant a standalone publication.
>
> We emphasize that our empirical results including the results with the 1B parameter model, and the SWARM experiments validate the effectiveness and practical usefulness of our method beyond doubt.
>
> - Lyu, Kaifeng, and Jian Li. "Gradient descent maximizes the margin of homogeneous neural networks." ICLR (2019).
> - Soudry, Daniel, et al. "The implicit bias of gradient descent on separable data." JMLR (2018)
> - Zhu, Zhihui, et al. "A geometric analysis of neural collapse with unconstrained features." NeurIPS (2021)
> - Wu, Yongtao, et al. "On the convergence of encoder-only shallow transformers." NeurIPS (2023).
> - Arora, Sanjeev, et al. "A convergence analysis of gradient descent for deep linear neural networks." ICLR (2018).
>
> ## Code
>
> As requested by the reviewer, we provide a minimal version of our code [here]. Upon publication, we intend to release the code for reproducibility and to enable future improvements.
>
> [here]: https://drive.google.com/file/d/10Alm3sgwSic3Hi3w7gbJrme5JgUcsP9j/view?usp=sharing
>
> ## Typo in Eq. 5
>
> Thanks for pointing out the extra closing parentheses. We will correct it.

---

> > ### Comment · Reviewer_xheX · 2025-04-04
> >
> > Thanks for the response. The theoretical results in this paper is generally valuable and the reviewer recommends to accept. However, due to the lack of the convergence analysis in stochastic scenario, the reviewer will maintain the rating.

---

> > > ### Author Response · Authors · 2025-04-05
> > >
> > > Thank you for acknowledging our response and recommending acceptance. We appreciate it!
> > >
> > > We agree with the reviewer that convergence analysis in the stochastic setting is an interesting research direction, which we intend to explore as a future work.

---

### Official Review · Reviewer_wivu · 2025-03-02

**Overall Recommendation:** 3

**Summary:**

This paper tells the story of overcoming a major challenge in training huge neural networks with pipeline parallelism. When models are split into stages running on different devices, asynchronous updates keep the pipeline full but introduce the problem of stale gradients, updates based on outdated information. To fix this, the authors reimagine Nesterov’s Accelerated Gradient method: they modify its look-ahead step to serve as a delay correction, effectively aligning the updates with the current gradient direction despite the delay. Their theoretical analysis shows that this approach converges reliably, and experiments on large-scale language models, even one with 1B parameters, demonstrate that it outperforms existing asynchronous methods and even beats synchronous training.

**Claims And Evidence:**

The claims are backed by both empirical and theoretical results.

**Essential References Not Discussed:**

n/a

**Experimental Designs Or Analyses:**

The experiments are conducted on several datasets and the proposed algorithm is compared to several previous works. The result include real-world scenario and convergence speed, both in terms of time in iterations. However, the proposed modification to NAG is not well justified empirically and the paper does not provide ablations for this modification.

**Methods And Evaluation Criteria:**

The evaluation includes several datasets for LLM evaluation.

**Other Comments Or Suggestions:**

n/a

**Other Strengths And Weaknesses:**

n/a

**Questions For Authors:**

* **Q.1.** How come the proposed asynchronous algorithm outperforms synchronous (GPipe) in terms of convergence speed in number iterations?
* **Q.2.** The modification to NAG can be see as a dependency between learning rate and momentum coefficient which might not hold in different settings. Would it make sense to validate this on a single worker setting? If not, then at how many workers does this modification start making senes?

**Relation To Broader Scientific Literature:**

This work extends the broader scientific literature on distributed optimization and parallel training by addressing the longstanding challenge of gradient staleness. Building on prior methods that either forecast gradients or use weight stashing and learning rate adjustments to mitigate delays, the paper uniquely repurposes the look-ahead step of Nesterov’s Accelerated Gradient method to serve as an intrinsic delay corrector. In doing so, it offers a theoretically sound and practically effective solution that outperforms earlier approaches in large-scale, asynchronous pipeline parallel training.

**Theoretical Claims:**

The proposed algorithm is backed by a theoretical convergence proof.

---

> ### Author Rebuttal · Authors · 2025-03-30
>
> We thank the reviewer for constructive feedback and address the specific concerns below.
>
> ## Ablations for NAG
>
> We have **already provided the ablation experiments for our modifications in Sec. 5.6**. Specifically, the effect of the momentum coefficient for NAG is reported in Fig. 6 and the effect of the gradient discounting term (our main modification to NAG in Eq. 9) is illustrated in Fig. 7. We believe we have clearly justified our modifications to NAG, however, if the reviewer believes any specific experiment is missing, we are happy to include them.
>
> ## How can an asynchronous method be better than GPipe?
>
> This is an intriguing question and we believe further study is required to completely understand the effects of asynchronous updates. Nevertheless, we answer this from a practical point of view below.
>
> There are two points to consider when comparing two algorithms: 1) the frequency of the weight updates, and 2) the effectiveness of the weight updates. Typically, asynchronous methods perform more frequent weight updates than synchronous methods (such as GPipe) when processing the **same amount of data**. Specifically, GPipe accumulates gradients for a particular number of steps (4 in our experiments) to increase the pipeline utilization (Huang et al., 2019)) and update the weights, whereas asynchronous methods update the weights for every microbatch. Since more frequent updates (if the updates are beneficial) translate into a faster reduction in loss, an asynchronous method can be better than GPipe.
>
> Here, the key point is “if the updates are beneficial” – as not all asynchronous methods are better than GPipe. In fact, Pipedream and Pipemare are significantly worse than GPipe despite doing 4 times more weight updates (Fig. 2). This takes us to the second point, the effectiveness of the updates. To test this, we trained our base model (134M model on WikiText, same as in the paper) with 2 pipeline stages and updated the weights at every microbatch for GPipe and compared against our method (same number of weight updates for both methods). Even in this case, **our method is better than GPipe**: training loss at 50k iterations is **3.275 vs 3.323**. This confirms that *weight updates of our method are more effective than GPipe updates*, and overall, it is an effective asynchronous PP optimization method.
>
> ## At how many workers does this modification start making sense?
>
> Since we consider pipeline parallel optimization, single worker setting does not make sense, and at minimum, we require 2 pipeline stages. In the paper, we have tested our method by varying the number of pipeline stages from 4 to 24 (Fig. 5). We now tested the same base model (134M model on WikiText) for 2 pipeline stages, and even in this case **our method outperforms GPipe in terms of both training and validation losses**:
>
> |Method|Train Loss| Validation Loss|
> |-|-|-|
> |GPipe|3.374|3.410|
> |Ours|**3.275**|**3.318**|
>
> This further confirms the effectiveness of our method even when the number of pipeline stages is small.
>
> ### NAG modification might not hold in different settings
>
> We agree with the intuition that our gradient discounting term can be seen as an interplay between learning rate and momentum coefficient. However, we are unclear about the reviewer’s point that “the modification might not hold in different settings”. In case our response above misses any setting that the reviewer is interested in, if the reviewer can be more specific, we can try to better clarify them.

---

> > ### Comment · Reviewer_wivu · 2025-04-01
> >
> > Thank you for the provided details, this partially satisfies my concerns. I'm still confused by how an asynchronous method outperforms a synchronous method in terms of better accuracy with respect to number of iterations. However, I accept your comment about the more frequent updates and therefore I will update my score.

---

> > > ### Author Response · Authors · 2025-04-02
> > >
> > > Thank you for acknowledging our response and updating the score. We appreciate it!
> > >
> > > This is one of the first works to test asynchronous PP methods in large-scale settings, and as you mentioned, the behaviour of asynchronous methods is not well understood. It would be an interesting research direction to theoretically/empirically understand how/when an asynchronous method can be better than synchronous methods.

---

### Official Review · Reviewer_tQWW · 2025-03-14

**Overall Recommendation:** 4

**Summary:**

The gradient staleness caused by existing asynchronous pipeline parallelism mechanisms hinders the practical usage in contrast to synchronous pipeline parallelims. This paper aims to solve tackle the stale gradients when using asynchronous pipeline parallelisms by nesterov accelerated gradients. The paper has made the following contributions: (1) the paper has demonstrated that the asynchoronous pipeline parallelism can outperform the synchronous alternative on large language model workloads with partical setups; and (2) the paper proposes both theoretical and empricial justification of the proposed methods.

**Claims And Evidence:**

While the paper has shown great accuracy improvement by the proposed methods, the performance evaluation is limited (Figure 5 right). It is thus obscure how the proposed methods will work in practice. The proposed methods should also be compared against other synchronous methods (e.g. PipeDream) in terms of performance to claim it is better than other synchronous methods.

**Essential References Not Discussed:**

I think the paper has already discussed all essential related works.

**Experimental Designs Or Analyses:**

While it is great to see the training curves of the proposed methods, it is obscure how the method could affect the generated outputs in practice. I would recommend the authors to add the inference results (e.g. the generated text from langauge models) of the trained models with different baselines.

**Methods And Evaluation Criteria:**

The paper is evaluated with commonly used language models and the experimental setups does align with the problem settings by comparing the loss curves of training against other baselines.

**Other Comments Or Suggestions:**

The paper's writing is in good shape. However, I would suggest the author to consider moving the related work section (section 4) as the second-to-last section of the paper.

**Other Strengths And Weaknesses:**

I do find the evaluation of the paper robust since it have evaluated on multiple widely used datasets. However, it would also be interesting to see how the proposed methods perform on other types of networks (e.g. convolutional networks).

**Questions For Authors:**

* How is the proposed method scalable & sensitive to multiple GPU servers?
* How much speedup can the proposed method achieve compared to its synchronous alternative?

**Relation To Broader Scientific Literature:**

The paper may also include the discussion about how the proposed methods could be applied to training with low-precision floating point numbers, since they are becoming more and more prevalent in training modern DNNs.

**Theoretical Claims:**

I do not find any problems in the theoretical analysis in the paper but my expertise is limited.

---

> ### Author Rebuttal · Authors · 2025-03-30
>
> We thank the reviewer for encouraging comments and address the specific points below.
>
> ## Scalability and sensitivity to multiple GPU servers
>
> Our **SWARM experiments were conducted on multiple GPU servers** (24 to be exact) in GCP, confirming that our method works seamlessly in such scenarios (Fig. 8 and Appendix B.1). We will include this information in the main paper. Additionally, in the paper, we have tested the scalability of our method by increasing the parameter count (Fig. 3) and by increasing the number of pipeline stages (Fig. 5). In both cases, our method **shows superior scalability**.
>
> ## Speed-up compared to its synchronous alternative
>
> Asynchronous methods offer 100% pipeline utilization by construction. In contrast, pipeline utilization of synchronous methods depends on the bubble size (Huang et al., 2019). For GPipe, the pipeline utilization is estimated as $\frac{N}{N+P-1}$ (Yang et al., 2021), where $N$ is the number of microbatches and $P$ is the number of pipeline stages. Typically, $N = P$ and therefore, the pipeline utilization of GPipe is about 50%. Based on this, **asynchronous methods would be twice as fast compared to GPipe** in a homogeneous environment with full GPU utilization. In a heterogenous, bandwidth constrained setup, the speed-up can be higher as asynchronous methods completely mask the communication bottleneck.
>
> ## Training time compared to synchronous methods
>
> As noted by the reviewer, we have **already provided the training time comparison in Fig. 5**, where it is clearly visible that *GPipe becomes exponentially slower* when increasing the number of stages compared to our method. The reviewer’s confusion might be due to misinterpreting Pipedream as a synchronous method. However, we would like to clarify that **Pipedream is an asynchronous method** and the *training time is the same for all asynchronous methods* including Ours, Pipedream, and Pipemare. The training curves for the 1B parameter model in Fig. 10 in the appendix illustrates this, and shows that the synchronous **GPipe is about 1.5x slower than the asynchronous methods** in this case.
>
> Nevertheless, as mentioned in the paper, the absolute wall-clock times are not comparable between asynchronous methods (ours, pipedream and pipemare) and the synchronous method (GPipe), due to the differences in the underlying implementation – asynchronous methods are based on the 6 years old third-party pipedream codebase, whereas GPipe is a pytorch official implementation.
>
> ## Inference results
>
> We have **provided perplexity scores on the validation set in Table 1** as a quantitative measure of inference results. Additionally, the validation loss curves were provided in Fig. 9 and also Fig. 3 right. As requested by the reviewer, we have now provided some generation results for GPipe and Our method below as qualitative results:
>
> |Prompt|GPipe Generation|Ours Generation|
> |-|-|-|
> |Cyclone|Iloʿ Trespass was a little slower organized than initially, but rapidly strengthened to become a hurricane … |Tropical Storm Bonnie (Harak 10) made landfall on the Outer Banks of North Carolina, dropping moderate rainfall … |
> |Largest country in the world|Hair bars at Festival Day Student Association events ...|France ...|
> |Wikipedia has a variety of articles in topics like|The regime constitutes a more moral dependence on Wikipedia than … |vernacular romances and classical history …|
> |Some of the popular sports in the world include|[Wurgar Canoe] (meaning football game) Wurgar Canoe, the city hall, … |Cricket and ice hockey (football, soccer, ice hockey, ice hockey, men's and women's hockey ) are the sports …|
>
> Note, this is the output of pretraining 1B model on the WikiText dataset which is relatively small and contrived to train useful LLMs compared to larger datasets like C4 or FineWebText. Furthermore, post-training plays a major role in incorporating instruction following capabilities. Nevertheless, both methods generate plausible English sentences relevant to the prompt, whereas our method seems to be better for some prompts. Also, as seen in Figs. 2, and 9, the models are not yet saturated and can be trained further to improve the generation quality.
>
> ## Convolutional networks
>
> To test the generality of our method, we trained a Resnet50 model on the TinyImageNet image classification dataset with 8 pipeline stages. In short, the conclusions in the paper hold:
> |Method|Top-1 Validation|
> |-|-|
> |GPipe|55.70|
> |Pipedream|55.02|
> |Ours|**55.79**|
>
> This is a small network and dataset in which Pipedream yields similar results to GPipe, even without any delay correction, highlighting that asynchrony is not a major pain point in these tasks. This observation aligns with previously reported results (Yang et al., 2021).
>
> ## Low-precision training
>
> We see no restrictions in using low-precision training in our method. We expect it to affect asynchronous and synchronous methods in the same way, however, experiments are needed to validate this.

---

> > ### Comment · Reviewer_tQWW · 2025-04-05
> >
> > Thank you for your response and the clarification provided. After reviewing the details, I have decided to maintain my original scores, as they already accurately reflect my evaluation.

---

> > > ### Author Response · Authors · 2025-04-06
> > >
> > > Thank you for acknowledging our response and recommending acceptance. We appreciate it!

---

### Decision · Program_Chairs · 2025-05-01

**Decision:**

Accept (poster)

**Comment:**

This paper presents addresses gradient staleness in asynchronous pipeline parallelism by reimagining Nesterov's Accelerated Gradient method. The authors modify the look-ahead step to serve as a delay correction mechanism, enabling reliable convergence despite asynchronous updates. The work provides both theoretical guarantees and empirical validation on large language models, including a 1B parameter model.

The paper has received positive reviews from all three reviewers, with particular praise for its theoretical foundations and practical impact. The authors have demonstrated that their asynchronous method can outperform synchronous alternatives in both convergence speed and final accuracy, while maintaining full pipeline utilization.

The authors have thoroughly addressed reviewers' questions during the rebuttal phase. They provided additional experimental results showing the method's effectiveness with as few as two pipeline stages, clarified the relationship between update frequency and convergence speed, and included qualitative results demonstrating the practical impact on model outputs. The authors also openly acknowledged limitations regarding theoretical analysis in stochastic settings, positioning this as future work.

Some limitations were noted, including the lack of theoretical analysis in the stochastic setting and questions about the precise mechanisms enabling asynchronous methods to outperform synchronous ones in terms of iteration efficiency. However, these limitations do not significantly detract from the paper's contributions.